# Ultrathin Photonic Polymer Gel Films Templated by Non-Close-Packed Monolayer Colloidal Crystals to Enhance Colorimetric Sensing

**DOI:** 10.3390/polym11030534

**Published:** 2019-03-21

**Authors:** Shimo Yu, Shun Dong, Xiuling Jiao, Cheng Li, Dairong Chen

**Affiliations:** National Engineering Research Center for Colloidal Materials and School of Chemistry and Chemical Engineering, Shandong University, Ji’nan 250100, China; 201411530@mail.sdu.edu.cn (S.Y.); shundong1995@mail.sdu.edu.cn (S.D.); jiaoxl@sdu.edu.cn (X.J.)

**Keywords:** polymer gel, colloidal crystals, optical film, pH sensor

## Abstract

Responsive polymer-based sensors have attracted considerable attention due to their ability to detect the presence of analytes and convert the detected signal into a physical and/or chemical change. High responsiveness, fast response speed, good linearity, strong stability, and small hysteresis are ideal, but to gain these properties at the same time remains challenging. This paper presents a facile and efficient method to improve the photonic sensing properties of polymeric gels by using non-close-packed monolayer colloidal crystals (ncp MCCs) as the template. Poly-(2-vinyl pyridine) (P2VP), a weak electrolyte, was selected to form the pH-responsive gel material, which was deposited onto ncp MCCs obtained by controlled O_2_ plasma etching of close-packed (cp) MCCs. The resultant ultrathin photonic polymer gel film (UPPGF) exhibited significant improvement in responsiveness and linearity towards pH sensing compared to those prepared using cp MCCs template, achieving fast visualized monitoring of pH changes with excellent cyclic stability and small hysteresis loop. The responsiveness and linearity were found to depend on the volume and filling fraction of the polymer gel. Based on a simple geometric model, we established that the volume increased first and then decreased with the decrease of template size, but the filling fraction increased all the time, which was verified by microscopy observations. Therefore, the responsiveness and linearity of UPPGF to pH can be improved by simply adjusting the etching time of oxygen plasma. The well-designed UPPGF is reliable for visualized monitoring of analytes and their concentrations, and can easily be combined in sensor arrays for more accurate detection.

## 1. Introduction

Developments in society have brought about renewed focus on environmental problems, and a growing need for simple and accurate sensors that can respond to environmental changes has emerged. In nature, there are various creatures that can smartly change color with the environment [1]. Chameleons, for example, can adjust the color of their skin to the color of their surroundings [2]. Zebrafish can change their appearance under light [3]. Some beetles respond to humidity. Inspired by this phenomenon, researchers have synthesized a series of materials that can respond to the surrounding environment by color.

Among various materials, responsive polymer gels are good candidates for mimicking the responsive colors in natures because they are able to change their volume by swelling or deswelling under the stimulation of external conditions and such change can be elaborately converted into optical signals to achieve a visual response to the analyte [4,5,6]. Based on this mechanism, several colorimetric sensors using polymer gels have been fabricated that respond to pH, temperature, humidity, glucose, macromolecules, and metal ions, among others [7,8,9].

Photonic crystals, a class of most interesting optical materials, are usually constructed by two or more media with different refractive indices arranged periodically in one, two, or three dimensions (1D, 2D or 3D, respectively) [10]. Because photonic crystals can produce band gaps for photons of a certain frequency, various film colors can be generated. Considering the unique advantages of photonic crystals in optical sensing, researchers have combined them with polymer gels to form advanced response materials. In a pioneer work, Asher’s group synthesized a polymer colloid array (PCCA) to quantitatively detect glucose by embedding polymer gel into 3D colloid crystals [11,12,13,14]; they also fabricated a 2D hydrogel sensor by attaching a 2D colloid array to hydrogel, and this sensor could change its lattice spacing by swelling of the gel to achieve a visual response to the analyte [15,16,17,18,19]. Polymeric 1D photonic crystals or so-called Bragg stacks have been fabricated by spin-coating of block copolymer solutions [20,21]. Despite their many benefits, however, the problem of slow responses greatly limits the development and application of these sensors. For example, the PCCA glucose sensor requires over 90 min to respond to the analyte [13].

To improve the response speed of sensors, a reverse opal polymer gel was developed using SiO_2_ or polystyrene (PS) microspheres as a template [20,21,22,23,24,25,26,27,28]. The presence of macropores in the film structure enhanced not only the responsiveness of the resulting sensor by increasing its specific surface area but also the transport of the analyte; the resulting response speeds were improved to a certain extent. However, 3D hydrogel pH sensors with an inverse opal structure still require a response time of 20 min to achieve equilibrium [23]. The response speed is proportional to the rate of volume change of polymer gels, and the change rate of gel volume is inversely proportional to the size of the gel [29]. An interfering gel film with a sub-micron size was synthesized by Zhang et al., and the complete response of this film to glucose was achieved within 2 min [30]. Li et al. fabricated reverse opal polymer gel thin films with sub-micron thickness by using MCCs as the template and achieved a fast response to pH within 1 min [31]. However, the stability of these films was weak and the linear relationship between the dip shift of the films and pH could not be maintained under the condition of strong acidity. Recently, we fabricated ultrathin polymer gel films that are infiltrated into MCCs, which show excellent stability in strongly acidic solution [32].

Besides response time, a good linear relationship between the stimulus and the response is of great significance to a sensor. An ideal sensor has not only a simple calibration process but also constant sensitivity and accuracy over the entire measurement range to enable reliable responses to the environment. However, research on the linear relationships of polymeric gel-based sensors is limited. 2D-PCCA films could respond to the analyte by changing the spacing of colloidal crystals, but a poor linear relationship among pH [15], antibiotics [16], and serum was found because the 2D structure of the colloidal array limited lateral swelling to some extent [17]. Although the interfering gel film shows a good linear relationship between the response and glucose concentration, it has very weak optical signals. Thus, the visual response could not be achieved [30].

Hysteresis is another common issue for polymeric gel-based optical sensors. The Donnan potential presents in low-ionic strength solutions and hinders protonation of gels by diffusion and thermodynamic exchange limit elimination of swelling, thus most polymer gel sensors show hysteresis [14]. Serpe’s group synthesized a PNIPAm-*co*-AAc microgel that could respond to pH and temperature [33]. The hysteresis loop size of this sensor could be adjusted by using solutions of various ionic strength and changing the concentration of AAc. Although the hysteresis was improved, the phenomenon remained; in theory, however, hysteresis may be completely eliminated after a long period. This problem seriously affects the practical applications of the film for continuous testing.

Targeted at above-mentioned issues, we present herein the fabrication of ultrathin photonic polymer gel films (UPPGF) by using non-close-packed monolayer colloidal crystals (ncp MCCs) as the template. Poly-(2-vinyl pyridine) (P2VP), a weak electrolyte, was chosen for this study because of its pH-dependent swelling ability. Swelling of the polymer gel led to changes in film thickness and shifts in reflection peak. The responsiveness of the proposed sensor was directly related to the volume change of the polymer gel. The filling fraction and total volume of P2VP in the film could be adjusted by controlling the packing density of the ncp MCCs by varying the time of O_2_ plasma etching, which was proven to play an important role in improving the responsiveness and linearity of the pH sensor. The film had an overall sub-micron thickness, which promoted ion diffusion and swelling of the gel, leading to fast response speed and small hysteresis loops. The good adhesion between the ncp MCC and the substrate enabled the ordered structure of the film to be maintained, ensuring good cycling stability of the sensor even after repeated testing. The well-designed UPPGF is simple and reliable for visualized monitoring of analytes and their concentrations, and can be easily combined in sensor arrays for more accurate detection by cross-sensing.

## 2. Materials and Methods

### 2.1. Raw Material and Reactants

Poly-(2-vinyl pyridine) (P2VP) (*M*_w_: 159 kg mol^−1^, Fluka), 1,4-diiodobutane (DIB) (99%, Alfa Aesar, UK), nitromethane (NM) (99%, Sinopharm, Shanghai, China), tetrahydrofuran (THF) (99.0%, Guangcheng, Tianjin, China), diethyl ether (DE) (99.5% Fuyu), styrene (95%, Beijing Chemical Co., Beijing, China, washed in NaOH before use), potassium persulfate (99.5%, Beijing Chemical Co., Beijing, China), sodium dodecyl sulfate (SDS) (*M*_w_: 288.38 kg mol^−1^, Kermel, Tianjin, China), ethanol (99.7%, Sinopharm, Shanghai, China), and ultrapure water (≥18.2 MΩ, Milli-Q Reference, Beijing, China) were used in this work.

### 2.2. Preparation of Close-Packed Monolayer Colloidal Crystals

First, polystyrene (PS) spheres (438 nm in diameter; standard deviation < 10%) were synthesized by standard emulsion-free polymerization [34]. Si wafers (one side polished) were cut into 1 cm × 1 cm squares, treated with piranha solution, rinsed with copious amounts of water, and dried under a N_2_ flow. The cp MCCs were then formed on the silicon wafer by the gas–liquid interface self-assembly method [35]. The adhesive force between the colloidal crystals on the silicon wafers was strengthened by annealing at 80 °C for 24 h. The cp MCCs were etched for different times (2, 4, 6, or 8 min) by O_2_ plasma etching with a 150 W power plasma cleaner (Beijing Huiguang Co., Beijing, China) to form ncp MCCs. The oxygen flow rate was maintained at 200 mL/min and 20 °C.

### 2.3. Preparation of UPPGF

Quaternization of P2VP was carried out following the procedure reported by Tokarev et al. [36]. With some modifications. Briefly, 0.1 g of P2VP and 0.1 mL of DIB were dissolved in a mixture of NM (4 mL) and THF (1 mL) under stirring at room temperature. Then, the solution was heated at 60 °C with stirring for 80 h to accelerate the quarternization reaction between P2VP and DIB. An excess amount of DE was added to the mixture, and centrifugation was performed to eliminate THF and the residual DIB. Then, qP2VP was dissolved in 5 mL of NM to form a 3.25 wt % solution for subsequent spin-coating. The qP2VP solution was spin-coated onto ncp MCCs at a speed of 1000 rpm for 1 min using a spin coater (Laurell-WS650). Finally, UPPGF was obtained after thermal crosslinking at 120 °C for 48 h.

### 2.4. Characterization

The morphology of the films was examined by a Hitachi SU8010 field emission scanning electron microscope (FE-SEM), and the reflection spectra of the samples were acquired with an Ocean Optics USB2000 fiber optic spectrophotometer coupled to a Leica DM2700 M optical microscope. The reflectance spectra were consistently measured from the same spot of a UPPGF specimen by saving the spot image to identify it in the following experiments. Optical micrographs were taken under white-light LED illumination by a Leica DFC450 digital color camera coupled to a microscope with a 10× objective lens.

### 2.5. Sensor Test

Solutions of a certain pH were prepared from 0.1 M citric acid (aq.) and 0.1 M trisodium citrate dihydrate (aq.), 0.05 M NaH_2_PO_4_ (aq.), 0.1 M Na_2_HPO_4_ (aq.), 0.05 M NaHCO_3_ (aq.), and 0.1 M NaOH (aq.). The pH of the buffer solution was measured by a pH meter (INESA PHSJ-3F). Each UPPGF sample was dipped into the buffer solution for 2 min and then blown using a N_2_ flow to eliminate the excess solution on its surface. The reflectance spectra of the UPPGF samples were recorded before and after the dipping process. The samples were recovered by soaking in pH 10 buffer solution for 2 s, washing with deionized water, and then blowing with N_2_.

## 3. Results and Discussion

### 3.1. Preparation and Characterization of UPPGF

The synthesis of ultrathin photonic polymer gel films (UPPGF) is shown in Scheme 1. We obtained close-packed (cp) monolayer colloidal crystals (MCCs) by gas–liquid interface self-assembly. As shown in Scheme 1a, the polystyrene (PS) microspheres were stacked in a dense hexagonal manner on the silicon substrate to form cp MCCs. The non-close-packed monolayer colloidal crystals (ncp MCCs-x) (x = etching time in minutes) was obtained by O_2_ plasma etching of cp MCCs. P2VP swells by protonation in acid solution.; thus, we selected P2VP as the responsive polymer in this study. The P2VP precursor solution (3.25 wt %, in NM) was immersed into the ncp MCCs by spin-coating (Scheme 1b). Finally, the P2VP was completely cross-linked with DIB in the vacuum drying oven at 120 °C to obtain UPPGF-x (x = etching time in minutes) (Scheme 1c).

The packing density of the ncp MCCs can be controlled by adjusting the O_2_ plasma etching time. The cp MCCs were treated at 80 °C for 12 h before etching to enhance the contact between PS and the silicon wafer and ensure that the film did not fall off or fold during etching and response detection. O_2_ plasma etching was then performed on the cp MCCs. Figure 1 reveals that the particle size of the PS microspheres gradually decreased with increasing etching time (from 438 nm for the sample without etching to 430 nm for ncp MCCs-2, 417 nm for ncp MCCs-4, 404 nm for ncp MCCs-6, and 395 nm for ncp MCCs-8). However, due to the good contact between PS and the silicon wafer, the position of the PS microspheres did not change during the etching process, and the gap between the microspheres increased gradually. Thus, ncp MCCs with different packing density were formed. Low-magnification SEM images reveal that the order of the array was not destroyed by plasma etching, and ordering of PS microspheres was maintained even after etching for 8 min. Such a characteristic is an important condition enabling films to display color and achieve a visual response to pH.

UPPGF was prepared by spin-coating a polymer precursor solution onto ncp MCCs and thermal crosslinking. The qP2VP was spin-coated onto the surface of ncp MCCs. As shown in Figure 2, the P2VP was coated uniformly on the surface of the PS microspheres but it did not completely fill the voids between spheres. The viscosity of P2VP is such that rapid spin-coating does not allow it to fully infiltrate the substrate structure. The thickness of the films decreased with increasing etching time (478 nm for qP2VP-infiltrated ncp MCCs-0 to 462 nm for qP2VP-infiltrated ncp MCCs-2, 440 nm for qP2VP-infiltrated ncp MCCs-4, 430 nm for qP2VP-infiltrated ncp MCCs-6, and 414 nm for qP2VP-infiltrated ncp MCCs-8) because the thickness of P2VP on the surface of the PS array was determined by the speed of spin-coating and the concentration of the precursor solution. The thickness of the films depended on the particle size of PS after etching when the speed of the spin-coating and concentration of qP2VP were held constant. Figure 2 shows that the array maintained its good order after thermal cross-linking. Although the temperature of thermal cross-linking was higher than the glass transition temperature of PS, the protective effect of P2VP prevented serious deformation of the microspheres. During thermal crosslinking, P2VP gradually infiltrated the gap between PS microspheres, which grew larger with increasing etching time and allowed more P2VP to infiltrate into the pores. Thus, waves were produced on the surface of the films. Compared with that before thermal cross-linking, the thickness of the films decreased (from 478 to 440 nm for UPPGF-0, from 462 to 419 nm for UPPGF-2, from 440 to 397 nm for UPPGF-4, from 430 to 390 nm for UPPGF-6, and from 414 to 387 nm for UPPGF-8). This finding is related to the slight deformation of PS microspheres and the infiltration of P2VP.

### 3.2. Optical Properties of UPPGF

To better understand the effect of O_2_ plasma etching on the structure of the film, we studied its optical properties. A high-refractive index silicon wafer (*n* ~ 3.5) was chosen as the substrate on which to construct UPPGF. Fabry–Pérot fringes are formed by reflecting the interference between the beams of the thin-film air and thin-film substrate interfaces [37]. Under normal conditions, the position of the interference peak wavelength conforms to Equation (1) [38]:(1)mλ=2nd,
where *n* is the refractive index of the film, *m* is an integer, and *d* is the thickness of the film. Calculations indicated that the peak of the cp MCCs was located in the visible region (585 nm; *d* = 438 nm, *m* = 2, and *n* = 1.335) [39]. In the experiments (Figure 3a), the center of the reflection peak was found at 588 nm. The valley observed was the result of multiple scattering from a single sphere, and the characteristic mode of 2D photonic crystals with hexagonal symmetry was found. The position of the valley gradually shifted toward shorter wavelengths with increasing etching time (625 nm for ncp MCCs-2, 617 nm for ncp MCCs-4, 611 nm for ncp MCCs-6, and 596 nm for ncp MCCs-8). This finding could be attributed to the refractive index of the film gradually decreasing with increasing etching time, because the distance between colloidal crystal microspheres did not change with the increase of etching time, but the ratio of air in the array increased, resulting in the decrease of effective refractive index of the film, consistent with the phenomena observed in the SEM images (Figure 1).

We constructed UPPGF by spin-coating and thermal crosslinking. We found only one reflection peak in the visible region after spin-coating of the qP2VP, and no valleys associated with the photonic characteristic mode were observed. This is because of the refractive indices contrast was eliminated when the P2VP was infiltrated into the films. In this case, the position of the reflection peak also moved toward shorter wavelengths with increasing etching time (688 nm for qP2VP-infiltrated ncp MCCs-0, 653 nm for qP2VP-infiltrated ncp MCCs-2, 619 nm for qP2VP-infiltrated ncp MCCs-4, 584 nm for qP2VP-infiltrated ncp MCCs-6, and 549 nm for qP2VP-infiltrated ncp MCCs-8; Figure 3b). Because the thickness of the film decreased gradually, the UPPGF obtained by thermal cross-linking showed the same trend (656 nm for UPPGF-0, 621 nm for UPPGF-2, 589 nm for UPPGF-4, 547 nm for UPPGF-6, and 533 nm for UPPGF-8; Figure 3c). The reflection peaks of all films demonstrated a certain blue-shift after thermal cross-linking, which was due to the decrease in film thickness. The above data are consistent with the change in film thickness observed in the SEM images (Figure 2).

### 3.3. Responsiveness of the UPPGF to pH and Mechanism Research

We tested the responsiveness of the UPPGF sensors to pH by immersing them in buffers of different pH. P2VP swells in acidic solution, and its degree of swelling is related to its degree of protonation. Figure 4 illustrates that the reflection peaks of the films did not change significantly when the films were immersed in alkaline solution (pH ≥ 7). In acidic solution (pH < 7), however, the reflection peaks of all films gradually shifted toward longer wavelengths with decreasing pH. Even under strongly acidic (pH = 2.57) conditions, this response was maintained because the PS array prevented the collapse of the film structure caused by the high degree of swelling. This phenomenon is shown more intuitively in Appendix A. As shown in Figure 4f and Appendix A, in comparison with that of UPPGF-0, the pH responsiveness of the UPPGF templated by non-close-packed monolayer colloidal crystals was improved, and UPPGF-4 and UPPGF-6 showed the best responsiveness to pH. The displacement of reflection peak of UPPGF-4 was 80 nm, about 30 nm longer than the wavelength shift of UPPGF-0. However, compared with that of UPPGF-6, the responsiveness of UPPGF-8 was reduced to a certain extent. It is well known that the sensing ability of polymer gel sensor is closely related to the volume of polymer gel. Thus, we think that the responsiveness of the UPPGF was determined by the volume of P2VP, and the volume change of P2VP in the UPPGF may be influenced by etching.

We further explained the variation of the pH responsiveness of the UPPGF with increasing etching time through simple calculation based on geometric model. In Scheme 2, we set the diameter of the PS microspheres as *D* and the total thickness of the film as *H*. The *D* of the PS microspheres decreased to *d* after etching, assuming that the thickness of P2VP on the PS microspheres *h* is unchanged. We then calculated the volume (*V_P2VP_*) of P2VP with decreasing *d*:(2)$$V_{P2VP}=V_{total}-V_{PS}=\frac{3\sqrt{3}D^{2}H-\pi d^{3}}{12}=\frac{3\sqrt{3}D^{2}(h+d)-\pi d^{3}}{12}$$
(3)$$V_{P2VP}^{\prime }=\frac{\sqrt{3}D^{2}-\pi d^{2}}{4}$$
(4)$$\Delta \lambda =\frac{2n\cdot \Delta H}{m}=\frac{2n\cdot V_{P2VP}\cdot f_{P2VP}}{m}$$

We found that the *V_P2VP_* did not always increase with decreasing *d*. Using Equations (2) and (3), we calculated that *V_P2VP_* increased gradually as *D* decreased from *D* to 0.74*D* but decreased gradually with further decreases in particle size beyond 0.74*D*, i.e., *V_P2VP_* reached its maximum value at 0.74*D*. According to Equation (4) (where Δ*λ* is the displacement of the reflection peak, Δ*H* is the change in thickness of the film, *n* is the refractive index of the film, *m* is an integer, and *f_P2VP_* is the swelling rate of P2VP), the displacement of the reflection peak is proportional to the variation in *V_P2VP_*. Therefore, the law of responsiveness of UPPGF to pH is well explained. In the actual tests, however, maximum volume was achieved even if the particle size was not reduced to 0.74*D*, likely because P2VP did not completely cover the film, as assumed, to form a smooth surface (Figure 2). Thus, *V_P2VP_* in the actual experiments decreased at a faster rate than predicted by the calculations.

We monitored the change in film thickness with decreasing pH through SEM to confirm our hypothesis. In Figure 5, the thickness of UPPGF-4 increased from 397 to 408, 419, 440, and 449 nm in response to immersion in pH 5.08, 4.20, 3.39, and 2.57 buffer solutions, respectively; the thickness of the UPPGF would still increase when pH is 2.57 with no structural collapse. Other films showed the same trend with decreasing pH (Appendix A). UPPGF-4 also showed the maximum thickness variation, which explains why UPPGF-4 had the best pH responsiveness.

### 3.4. Linearity of the UPPGF to pH and Mechanism Research

Sensors with good linearity can detect analytes more accurately in the whole range of measurement. As shown in Figure 6a, we found that the linearity towards pH was gradually enhanced with increasing etching time. The coefficient of determination (R^2^) of the sensor gradually increased from 0.95684 (UPPGF-0) to 0.9816 (UPPGF-8). This phenomenon can also be explained by Scheme 2. The relationship between the filling fraction of P2VP (*N_P2VP_*) and PS particle size (*d*) is in accordance with Equation (5):(5)$$N_{P2VP}=1-\frac{\pi d^{3}}{3\sqrt{3}D^{2}(h+d)}$$

Equation (5) reveals that *N_P2VP_* always increase with decreasing *d*. We thus considered that *N_P2VP_* is an important factor affecting the linearity of response. Although the PS array provided the necessary optical signals for the film, it could restrain the regular swelling of the polymer gel. Therefore, the linearity of UPPGF increased with increasing etching time. Considering response degrees and linearity of response, UPPGF-6 exhibited the best properties among the synthesized films.

### 3.5. Response Speed of the UPPGF to pH

Fast response speeds are widely favored in practical applications. We measured the response speed of the UPPGF by immersing them in buffer solution of pH = 3.39 and recording the change in reflection peak over time. Approximately 90% of the total response could be achieved within 10 s by the films, and stable responses could be achieved within 2 min (Figure 6b). The response speed of UPPGF was faster than that of the inverse opal hydrogel pH sensor (20 min) reported by Braun et al. [23] and the 2D-PCCA pH sensor (30 min) reported by Asher et al. [15], and comparable with our previously reported inverse opal monolayers of P2VP gels [31] and ultrathin P2VP gel-infiltrated MCCs films [32]. This fast response speed was due to the structural characteristics of the UPPGF, which included submicron thickness. The response speeds of UPPGF did not decrease with increasing volume of P2VP. Although the volume of P2VP increased after etching, the thickness of the whole film decreased.

### 3.6. Stability of the UPPGF to pH

We tested the stability of the UPPGF samples. Folding or shedding was not found in the SEM images when the UPPGF responds to the pH buffer solution (Figure 5 and Appendix A). Then, the films were immersed in buffer solution of pH = 3.39 and then recovered in a solution of pH = 9.17. Figure 6c reveals that the position of the reflection peak remains basically unchanged after 10 cycles, thereby indicating good cyclic stability. Although more P2VP was in contact with the substrate with increasing etching time, the contact force between PS and the silicon substrate was improved through heat treatment of the cp MCCs at 80 °C prior to etching. Thus, the UPPGF sensors showed good stability.

### 3.7. Hysteresis Loops of the UPPGF to pH

Sensors with low hysteresis loops can accurately detect the environment regardless of their input history. The peak shifts of the UPPGF samples observed under two approaches of pH input, i.e., from pH 9 to 2 and from pH 2 to 9, were recorded, and film recovery was found to be unnecessary for detecting different pH solution. As shown in Figure 6d, all UPPGF exhibited small hysteresis loops with short test times (2 min), likely because of the submicron thickness of the film. Ions could diffuse rapidly through the film, and the polymer gel could swell and shrink quickly. Thus, UPPGF can be used to test the environment continuously.

### 3.8. Visual Response of the UPPGF to pH

Importantly, the UPPGF achieved visual response to pH. In Figure 7, we can see that the color of the film changed with the change of pH. Compared with UPPGF-0, the color change of the UPPGF templated by ncp MCCs was more obvious. This phenomenon is consistent with the change of the redshift of the reflection peak with the etching time (Figure 4). Moreover, other responsive materials (for example, other responsive polymer gels, MOF, etc.) can be combined with ncp MCCs to improve their sensing capabilities. A sensor array could be obtained by the combination of the films etched at different times to achieve more accurate analysis.

## 4. Conclusions

In summary, UPPGF using non-close-packed monolayer colloidal crystals (ncp MCCs) as template were prepared by spin-coating of qP2VP and subsequent thermal cross-linking. The UPPGF showed bright structural colors depending on the thickness of the film. The swelling of P2VP gel in acidic solution caused changes in the film thickness, thus resulting in a change in the visible color. Compared with those templated by cp MCCs, the UPPGF not only increased the wavelength shift of the reflection peak upon pH changes, but also improved the linearity of response, while maintaining the advantages of fast response speed, high cycle stability and small hysteresis loop. The packing density of ncp MCCs could be adjusted by controlling the time of oxygen plasma etching to effectively regulating the volume and filling factor of P2VP, which could be used to improve the responsiveness and linearity. The present strategy of fine-tuning the volume of filling fraction of polymeric gel in a sensor device can be feasibly extended to other responsive materials/systems to improve the responsiveness and linearity, as well as makes it simple to construct sensor arrays to enable more accurate detections.

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
