# Peer review of "Ultrathin Photonic Polymer Gel Films Templated by Non-Close-Packed Monolayer Colloidal Crystals to Enhance Colorimetric Sensing"

_polymers, 2019, doi:10.3390/polym11030534_

Round 1
Reviewer 1 Report
Ultrathin photonic polymer gel films templated by non-close-packed monolayer colloidal crystals for enhanced colorimetric sensing
Shimo Yu, Shun Dong, Xiuling Jiao, Cheng Li, and Dairong Chen
Herewith I am submitting my reviewer comments for the above-mentioned manuscript which is under considerations to be published in polymers. The article is about colorimetric sensing using polymer spheres in a regular sub-monolayer as template. The idea is very interesting and nicely done in the article. Although there is already quite some literature about such responsive materials, the authors sufficiently explain how their contribution fits into the existing literature and where they improve. Their material responds to changes in pH by changing its color. Overall the article is well written, the English is of sufficient quality and the figures are optically appealing. Overall I believe that the manuscript is suitable for publication in Polymers. Below are a few minor comments.
Line 52: There is something grammatically wrong in the sentence starting with: “Asher’s group has synthesized pioneeringly”
Line 110: “Materials Poly-(2-vinyl pyridine) (P2VP)” I would remove the word materials in the beginning
Line 120: ” and dried under a N2 flow” the 2 should be lower case (also several times from line 143)
Line 123: “by O2 plasma etching with a” the 2 should be lower case (also several times below)
Line 150: “with N2.a N2 flow to” There is a space missing. The A after the point should be capitalized
Line 229: There is something grammatically wrong in the sentence starting with: “This is because that the refractive indices”
Fig 4: There are no error bars in Fig f. Also the inset f) is not labeled (it only goes from a-e)
Line 275: “faster rate than that predicted by the” should be “faster rate than predicted by the”
Line 290: “From Fig. 6a,we can find that the linearity towards pH was gradually enhanced with increasing etching time.” I don’t think one can say the linearity enhances. In this case there is just not a linear dependency between pH and the signal (which is perfectly fine)
Author Response
Thank you for you revision of our manuscript (polymers-407789). We are very grateful for the referees’ constructive suggestions and comments which are very helpful for improving the manuscript. According to the referees’ suggestions, we have carefully revised the manuscript and supporting information.
The revised manuscript and Supporting Information etc. have been uploaded. With the above revisions (marked with red color in the manuscript), we hope it can meet the requirement for publication. All the authors have approved the revised manuscript. If you have any queries, please feel free to contact us.
Sincerely yours,
Cheng Li

Reviewer 2 Report
In this work, a facile and effective method was introduced to improve the photonic sensing properties of polymeric gels by using non-close-packed monolayer colloidal crystals. The ultrathin photonic polymer gel film show improvement in responsiveness and linearity towards pH sensing. I found this work has interesting results. However, several major issues have to be taken into account and address for this manuscript. Few main concerns are considered as follows:
1- Title is not meaningfully describing the aim of this work. It is recommended to revise that by adding To enhance instead of For enhanced. This is just a recommendation.
2- Several typos can be seen. for example, in line 147, (..then blowing with N2.a N2..). In another example, line149-151 as ( The samples were recovered by soaking in pH 10 buffer solution for 2 s, washing with deionized water, and then blowing with N2.a N2 flow to eliminate the excess solution on its surface.) is similar to line 146-148.! Line 152-153 is equivalent to line 146-147. Thus, carful revision is highly needed.
3- Authors used ultrathin photonic polymer gel films using an abbreviation as UPPGF in line 156, but in line 211, 330, and 340 used the whole name.! If authors use an abbreviation, then they should use that. In the same way, for Equation 1 needs a citation.
4- English correction is highly recommended. For example, in line 216 starts with: calculations with using a capital letter.!
5- As shown in Fig.3, the blue-shift tuning wavelengths was observed by increasing etching time. Authors expressed that is due to decreasing the size of the microspheres. Actually, the relationship of the size of microspheres with the tuning reflection shift is not an obvious reason here. Authors explanation can illuminate this vague.
6- Figure 4 needs an extra label as Figure 4(f) for Δλ. Moreover, authors should describe this finding in the main text as well.
7- Equations 2-4, don’t have the citation to determine the variation of the pH responsiveness. The same story for Equation 5. Are they new proposed Equations by authors? Or obtain from literature?
8- In line 306-307, authors compare their results with Braun et al and Asher et al without mentioning their citations. Authors should add suitable citations for these references.
9- Scale bar for Figure 7 is absent.
Author Response
Thank you for your reviesion of our manuscript. We are very grateful for the referees’ constructive suggestions and comments which are very helpful for improving the manuscript. According to the referees’ suggestions, we have carefully revised the manuscript and supporting information.
The revised manuscript and Supporting Information etc. have been uploaded. With the above revisions (marked with red color in the manuscript), we hope it can meet the requirement for publication. All the authors have approved the revised manuscript. If you have any queries, please feel free to contact us.
Sincerely yours,
Cheng Li

Round 2
Reviewer 2 Report
The authors have adequately addressed my concerns with the paper and I now recommend it for
publication.